# Human Operation Augmentation through Wearable Robotic Limb Integrated with Mixed Reality Device

**DOI:** 10.3390/biomimetics8060479

**Published:** 2023-10-08

**Authors:** Hongwei Jing, Tianjiao Zheng, Qinghua Zhang, Kerui Sun, Lele Li, Mingzhu Lai, Jie Zhao, Yanhe Zhu

**Affiliations:** 1State Key Laboratory of Robotics and System, Harbin Institute of Technology, Harbin 150001, China; jhw_hit@163.com (H.J.); zhengtj@hit.edu.cn (T.Z.); 19b908069@stu.hit.edu.cn (Q.Z.); 21B908043@stu.hit.edu.cn (K.S.); 21S108246@stu.hit.edu.cn (L.L.); jzhao@hit.edu.cn (J.Z.); 2School of Mathematics and Statistics, Hainan Normal University, Haikou 571158, China; laimingzhu@126.com

**Keywords:** human augmentation, wearable robotic limb, supernumerary robotic limb, mixed reality, human–robot interaction

## Abstract

Mixed reality technology can give humans an intuitive visual experience, and combined with the multi-source information of the human body, it can provide a comfortable human–robot interaction experience. This paper applies a mixed reality device (Hololens2) to provide interactive communication between the wearer and the wearable robotic limb (supernumerary robotic limb, SRL). Hololens2 can obtain human body information, including eye gaze, hand gestures, voice input, etc. It can also provide feedback information to the wearer through augmented reality and audio output, which is the communication bridge needed in human–robot interaction. Implementing a wearable robotic arm integrated with HoloLens2 is proposed to augment the wearer’s capabilities. Taking two typical practical tasks of cable installation and electrical connector soldering in aircraft manufacturing as examples, the task models and interaction scheme are designed. Finally, human augmentation is evaluated in terms of task completion time statistics.

## 1. Introduction

The supernumerary robotic limb (SRL) is a new type of wearable robot that is different from prosthetic and exoskeleton robots. It improves the human body’s ability to move, perceive, and operate through mechanical and human limbs’ integration, mutual assistance, and cooperation [1,2,3]. According to different application scenarios, scholars from various countries have designed SRLs that can be worn in different positions. Specifically, it includes an extra robotic limb worn on the waist to complete auxiliary support tasks [4] or tasks of tool delivery and remote assistance [5]. Extra robotic limbs are worn on the shoulders to complete overhead support tasks [6]. Extra robotic fingers are used to improve the independent living ability of the disabled [7] or to expand the grasping capacity of ordinary people [8]. In these application scenarios, the SRLs play a vital role in expanding the ability of a single person to operate and improve work efficiency.

It is also imperative to design appropriate interaction methods to make SRLs not affect the wearer’s operation ability and rationally use human body information. Ref. [9] divided the command interfaces of the supernumerary effector into three categories: body, muscle, and neural. The body interface refers to the interaction method based on body motion mapping, such as fingers [10], feet [11], and the interaction method using changes in hand and finger force [12]. Surface electromyography (EMG) interaction methods based on redundant human muscles are proposed, such as chest and abdominal EMG signals [13] and forehead EMG signals [14]. The electroencephalography/magnetoencephalography (EEG/MEG) interaction method [15,16] has also been proposed but not applied to actual task scenarios. Although these interaction methods can realize the essential communication of SRLs, there are still some unsolved problems, such as the increased cognitive load of people, complex sensor data processing, and limited adaptability (wearer adaptability and application scene adaptability). Therefore, it is significant to construct a simple, reliable, and natural human–robot interaction interface for SRLs and to design an efficient interaction strategy.

The use of collaborative robots in the industry has grown over the past few years. Augmented Reality (AR), as a prominent and promising tool to support human operators in understanding and interacting with robots, has been applied in many human–robot collaboration and cooperative industrial applications. Research on the human–robot collaboration of head-mounted displays (HMDs) and projectors has attracted more attention [17]. Compared with industrial collaborative robots, SRLs have the characteristics of closer human–robot interaction, moving with the wearer and not affecting the wearer’s operation ability. Mixed Reality (MR) is a broader term encompassing AR and Augmented Virtuality (AV). It is a general term for blending the virtual and the physical worlds [18,19]. HoloLens2 is an MR device developed by Microsoft Corp. Unlike most other such devices, it is a complete MR system running the Windows 10 operating system and containing a central processing unit, custom-designed holographic processing units, various types of sensors, see-through optics with holographic projectors, etc. Therefore, the main focus of this work is to apply an integrated MR device to call various information about the human body to realize the enhancement of the human body operation by the SRLs, as illustrated in Figure 1. Through quantitative methods, the augmentation of the human body by the SRL system integrated with HoloLens2 is evaluated.

The structure of this paper is as follows. First, the research work related to this paper is introduced. Then, the composition of the system and the interaction framework combined with HoloLens2 multi-source information are explained to realize the interaction with SRL. Aiming at cable installation and electrical connector soldering tasks, the task models and interaction methods are constructed. Finally, through comparative experiments, the feasibility of the system and the enhanced performance on the human body are evaluated.

## 2. Related Work

### 2.1. Human–Robot Interaction Method of SRLs

Unlike traditional collaborative robots, SRLs have a closer human–computer interaction mode and a cooperative mode of moving with the human body. Therefore, the current human–robot interaction methods for SRLs focus on researching natural interaction methods that do not affect the wearer’s ability to operate. Refs. [12,13] have proposed an interaction method based on task redundant finger strength to complete the task of opening the door and an interaction method based on EMG signals. Ref. [20] proposed an interaction method based on eye gaze information and gave quantified manipulation accuracy. Ref. [21] proposed a method for assisted walking foot position prediction that fuses continuous 3D gaze and environmental background. Combining human gaze and environmental point cloud information is significant for studying wearable robotic limb control for assisted walking. Ref. [22] proposed a new gaze-based natural human–robot interaction method that suppresses the noise of the gaze signal to extract human grasping intent. This study is different from the previous research on gaze intention, and it has reference significance for the study of human–robot natural interaction with wearable robotic limbs. Ref. [15] researched the method of manipulating external limbs by EEG signals and evaluated the influence of various factors of SRLs on human–robot interaction. Ref. [8] analyzed the ability to control extra robotic fingers with the toes and the effects on human nerves. Ref. [23] proposed a task model for overhead tasks to realize the human–robot collaboration of SRL according to the operator’s actions. This task model has the adaptability of the task and has reference significance for the construction of interaction strategies for similar tasks. The interaction between SRLs and the wearer mainly converts the wearer’s intentions into robot task execution information. This can be understood as a “decomposition” and “synthesis” process. People decompose their intentions into various external data, and the SRLs collect the external information to synthesize the task execution information required by the robot. The essential lies in the collection of human multimodal information, the synthesis and transformation of multi-source information, the reduction in human cognitive load, and the strategy of SRL cooperation. The existing work has no precedent for applying MR to the human–robot interaction of SRLs. Moreover, most current interaction methods call for a single interaction method, and the user experience is not comfortable enough. Here, our work is to apply an integrated device that can obtain multimodal information about the human body to realize the natural manipulation of the SRL and the information feedback through MR. The interaction method is evaluated with two practical application tasks and qualitative and quantitative analysis.

### 2.2. AR-Based Operator Support Systems

Advances in display and vision technologies create new interaction methods that enable information-rich, real-time communication in shared workspaces. The visualization methods applied by AR in human–computer collaboration mainly include HMDs, spatial augmented reality projectors, fixed screens, and hand-held displays (HHDs) [17]. The current research trend mainly implements Safety, Guidance, Feedback, Programming, and Quality Control through HMDs or projectors. Ref. [24] communicate the motion intent of a robotic arm through a Mixed Reality head-mounted display. Ref. [25] applied AR in the human–robot collaborative manufacturing process to solve human–robot safety issues and improve operational efficiency. Ref. [26] present the design and implementation of an AR tool that provides production and process-related feedback information and enhances the operator’s immersion in safety mechanisms. Refs. [27,28] also researched the application of AR to realize industrial robot programming and collaboration models. These research results have shown the advantages of AR devices in human–robot interaction through qualitative or quantitative methods. In this paper, we mainly research the control of the wearable robotic arm and the feedback on the robot through HoloLens2. The goal is to enhance the operator’s operating ability, relieve the operator’s psychological pressure, and improve operating efficiency. We summarized and analyzed the work related to this paper, as shown in Table 1.

## 3. Methods

### 3.1. System Composition

The wearable robotic arm system comprises a 6-degree-of-freedom (6 DOF) SRL previously introduced in our prior work [29], which leverages the wire rope drive mechanism, an Intel RealSense T265 (Intel, Santa Clara, CA, USA) camera and the HoloLens2, as illustrated in Figure 2. The SRL system encompasses a comprehensive 6 DOF robotic arm, the UP board development board operating on the Ubuntu 18.04 system, various drivers, and controllers. In the drive system of the SRL, we employed steel wire with a diameter of 1.37 mm (7 × 49 construction, breaking strength of 1130 N) manufactured by the German Carl Stahl Company. This steel wire comprises 343 strands, offering exceptional flexibility compared to conventional 7 × 19 and 7 × 7 wire ropes. The three rotational degrees of freedom in the shoulder and elbow joints are driven by AK80-9 actuators (230 W, 9 N.m, 485 g) mounted near the rear. The wrist joint, featuring three degrees of freedom, is directly actuated by Dynamixel XH430-W250 (3.4 N.m, 82 g) and 2HL430-W250 (1.4 N.m, 98.2 g). The two-fingered gripper is powered by Dynamixel XH430-W250 (3.4 N.m, 82 g) and is constructed from 3D-printed resin material.

To reduce weight and enhance structural strength, the arms are fabricated from standard carbon fiber tubes with inner diameters of 26 mm and outer diameters of 30 mm. Most other components are constructed from aluminum alloy (AL6061-T6). The total weight of one robotic arm, including its structure, motors, and sensors, amounts to 3.45 kg. The HoloLens2 headset weighs 566 g, while additional components such as the backplate, controller, and battery contribute an extra 2.2 kg. Consequently, the overall system weight totals about 9.67 kg.

To ensure wearer comfort, we designed a backplate that conforms closely to the wearer’s back and secured it with flexible fabric straps. We simplified the wearing process by carrying the SRL robot on the back like a backpack, with adjustable buckles for tension control. The HoloLens headset is worn directly on the head and can be adjusted for fit using a rear-mounted knob. Safety measures have been implemented in the SRL, including mechanical limits and software-based safeguards, enhancing the wearer’s security to a certain extent. The overall system’s wearing configuration is convenient, but its weight, though not excessive, may potentially cause some discomfort during extended use.

The working mode of the wearable robotic arm is to move with the body and cooperate closely with the wearer. Therefore, we want to manipulate the robot and obtain effective feedback without affecting the human’s ability to operate. As a head-mounted display, HoloLens2 is easy to wear and intuitive to use. HoloLens2 can more conveniently obtain multi-modal information about the human body and provide rich feedback to realize an efficient and natural interaction mode. In this process, HoloLens2 is a communication medium between the wearer and the SRLs, as shown in Figure 2.

The program construction of HoloLens2 and the integration method of SRL are shown in Figure 3. The system consists of 5 modules: computer development module, HoloLens2 module, Up board-based control system, SRL, and wearer. The computer development module refers to the program development work of HoloLens2. To realize the AR model presentation and real-time feedback of SRL, the SRL model (.FBX) was exported through 3D model software and imported into the Unity development platform. Then, we combined the Mixed Reality Toolkit and C script with building program functions and scenes. The expanded program can be tested in the simulator in advance and released on HoloLens2 after passing. According to the characteristics of the close connection between the wearable robotic arm and the wearer, the interactive information that HoloLens2 can provide includes eye gaze, head and hand gesture information, spatial awareness, voice input and output information, and AR object information. Through the call of multimodal information, the wearer can interact with HoloLens2. HoloLens2 and Up board realize message communication through a wireless network (TCP/UDP) and control SRL to cooperate with the wearer through CAN communication. The delay of the communication process is within 20 ms, which is enough to meet the basic needs of the wearer.

### 3.2. Interaction Framework Based on Mixed Reality

Building an application through which the wearer interacts with the SRL is a complex process based on Mixed Reality. This mainly depends on the interactive information that the wearer can express in the working situation. To propose a complete interaction framework, the way of information transfer and fusion in the process of SRL augmentation assisting the wearer needs to be analyzed. The role of HoloLens2 mainly includes the transmission of the wearer’s intention (task information), the working mode of SRLs and the feedback of sensor information, the operation guidance and assistance of the wearer’s operation, and the safety protection of the wearer. Specifically, the transmission of human intentions can also be decomposed into the target location, task content (specified task operation), and task trigger timing. The trigger timing of the task is critical, which involves the rationality of the SRL movement timing. For example, Leader–Follower [10,30] is a typical interaction tactic that needs to set the timing of human–robot cooperation. The feedback information includes the feedback information sent by the SRL to the wearer and the feedback control signal sent by the wearer to the SRL. Operation guidance information can include operation prompts, object marking prompts, operating strength displays, dimensions instead of physical measuring instruments, etc.

The interaction architecture focuses on effectively combining the human body’s multimodal information with the functional modules of Mixed Reality, as shown in Figure 4. The picture also shows some actual application screens of Mixed Reality. Human body multimodal information mainly includes body gesture tracking, eye gaze, and speech input. The Mixed Reality module mainly includes spatial awareness, virtual display, AR objects, and stereo audio. Facing different character scenes, the cognitive load of each part of the wearer’s body is different. Therefore, the wearer’s intention can be extracted by combining different modality information and Mixed Reality. For example, the gaze is designated as a ray with direction, which can be combined with spatial awareness to directly obtain the target position intention in real space. The time spent gazing at the AR object can be used as a criterion for triggering timing. Head and hand gestures can be combined to define different categories as in other studies directly. They can also be combined with AR objects to begin predetermined operations and even manipulate the movement of SRL. Speech input can intuitively describe task operation information and trigger timing. Speech controls are adequate when tasks occupy other parts. These are the intentions communicated to the SRL by the wearer. SRL can give visual and auditory feedback to the wearer through virtual presentation and stereo sound. Therefore, a suitable combination of multimodal information and Mixed Reality is sufficient to achieve natural, low-brain-load manipulation and perception of SRL.

### 3.3. Real-Time Tracking Method

To construct the interaction method between HoloLens2 and the SRL and share the virtual space data, we propose a method to establish the relationship between the virtual space coordinate system and the actual space coordinate system. Regarding tracking methods, there are mainly marker-based and markerless approaches, which are gradually increasing. Because of the projection system, the feature extraction software focuses more on tracking the target. However, the markerless approach generally has higher requirements on the performance of the equipment and a large amount of calculation. Thus, we use a marker-based method for tracking. Although some may use the markers for the first calibration, once completed, they require little recalibration since they are not removable. The device moves with the body for a wearable robotic arm, so a real-time tracking method is needed. Figure 5 shows the coordinate system positions between the overall system components.

To establish the relationship between the virtual space coordinate system and the robot real space coordinate system, we designed a coordinate transfer method using the Vuforia QR code recognition technology and the position and attitude of the “virtual object” presented by AR as an intermediate medium. Since the position of the SRL moves with the wearer, the T265 tracking camera is used to obtain the pose change in the robotic limb base, and the relationship between the origin of the virtual space coordinate system and the robotic limb real space coordinate system is corrected in real time. The specific calculation method is constructed as follows.

The variables determined by the installation position are defined as follows: the transformation matrix of the QR code relative to the SRL is $T_{Mark}^{Base}$, and the transformation matrix of the T265 camera relative to the SRL is $T_{T265}^{Base}$. When the tracking relationship is established for the first time through the Vuforia recognition function, the transformation matrix of the QR code relative to the origin of the virtual space coordinates is obtained as $T_{Mark}^{Holo}$. It can be deduced that the transformation matrix of the origin of the virtual space of HoloLens relative to the base of the SRL is $T_{Holo}^{Base}$. Because the SRL base moves with the wearer, $T_{Holo}^{Base}$ needs to be constantly updated. By accessing the T265 camera data, the attitude transformation matrix of the T265 camera, $T_{T265Now}^{T265Old}$, is acquired in real time. The real-time coordinate transformation relationship between SRL in real space and HoloLens virtual space is derived as $T_{Holo}^{BaseRT}$.
(1)$$T_{Holo}^{Base}=T_{Mark}^{Base}\cdot T_{Holo}^{Mark}$$
(2)$$T_{Holo}^{BaseRT}=T_{T265}^{Base}\cdot {\left[T_{T265Now}^{T265Old}\right]}^{-1}\cdot {\left[T_{T265}^{Base}\right]}^{-1}\cdot T_{Holo}^{Base}$$

Therefore, through the above coordinate system relationship construction method, we can communicate the AR object information in HoloLens with the SRL to realize the application of multimodal communication.

### 3.4. Task Model

#### 3.4.1. *FSM* Task Modeling

The wearer’s collaboration with the SRL can be divided into different task phases. The task model includes task state, parameters, and switching conditions. The task model can also be fine-tuned to adapt to similar process tasks. We first build task models for the cable installation and electrical connector soldering tasks.

In the aircraft manufacturing process, workers need to install many cables in the cabin, and the problems they face are the neat arrangement of cables and the bundling of multi-strand cables. Workers need to grasp and tension the cables with both hands to complete the bundling task, which is complex and cumbersome for both hands. The task of soldering cables and electrical connectors is equally challenging. It involves the manipulation of two soldering objects, a soldering tin and a soldering iron: a total of four things. If the SRL can assist workers in completing these tasks, it will improve the worker’s operational efficiency and comfort.

The process of SRL assisting the operator to complete the cable installation task can be expressed as follows: (i) the operator organizes the cables and places them near the installation location; (ii) the SRL helps the operator grasp the cables and apply a specific force to keep the tension of the cables; (iii) the operator concentrates on installing and fixing the cables; (iv) after the cables are installed, the SRL returns to the initial position. The process of the SRL assisting the operator to complete the soldering task is similar but more complicated: (i) the SRL grabs the electrical connector delivered by the operator; (ii) the operator drags the end of the SRL to the vicinity of the position to be soldered; (iii) the SRL adjusts the posture of the electrical connector according to the posture of the soldering iron, giving the operator a comfortable soldering angle; (iv) the operator completes the soldering of the electrical connector and cable; (v) after the task is completed, the SRL returns to the initial position.

Different task states can be divided according to the flow of the task. The cable installation task and electrical connector soldering task model based on *FSM* can be expressed as a tuple containing the following: task state space *S*, state switching condition *T*, SRLs motion mode set *M*, and state transition function σ($S_{i}$, $T_{i+1}$):(3)FSM=<S,T,M,σ>

Figure 6 and Figure 7 are schematic task flow diagrams of the cable installation and electrical connector soldering tasks. The picture in the figure is the first-person perspective of the simulated operation in HoloLens. We use virtual cylinder lines to represent analog cables, and mark points are set at equidistant positions in the middle of the cylinder to guide the cable-tying operation, as shown in Figure 6. The model of the SRL is imported to enable it to realize an inverse kinematics solution and clamp the cables. In the virtual scene of the soldering task, we adopt the same method to represent the cable with a cylinder model and import the SRL model into the scene, as shown in Figure 7. The operator drags the end of the SRL to the vicinity of the red cable. HoloLens2 recognizes and tracks the soldering iron’s posture, and AR technology generates a recognition square in the field of vision. Then, the SRL adjusts the posture to adapt to the soldering posture. When the operator completes the soldering, the red virtual cable turns green.

As shown in Figure 6 and Figure 7, the cable installation operation process includes four states: “preparation state” (*PS*), “cooperation state” (*CS*), “installation state” (*IS*), and “end state” (*ES*). The operation process of the soldering task includes five states: “preparation state” (*PS*), “drag state” (*DS*), “cooperation state” (*CS*), “soldering state” (*SS*), and “end state” (*ES*). The state space of the cable installation task ($S^{I}$) and soldering task ($S^{W}$) can be expressed as follows:(4)$$\left\{\begin{array}{c} S^{I}=\left\{S_{0}^{I},S_{1}^{I},S_{2}^{I},S_{3}^{I}\right\}=\left\{PS,CS,IS,ES\right\}\\ S^{W}=\left\{S_{0}^{W},S_{1}^{W},S_{2}^{W},S_{3}^{W},S_{4}^{W}\right\}=\left\{PS,DS,CS,SS,ES\right\} \end{array}\right.$$

The switching condition *T* of the *FSM* task model is used to realize the transition of the task state. The switching condition *T* is jointly determined by the operator’s intention information and the sensor information of the robot. When the switching condition *T* is triggered, the SRL executes the motion *M*. The task state *S* switches from the previous state to the next one. The switching condition *T* and the motion mode *M* can be expressed as follows:(5)$$\left\{\begin{array}{c} T^{I}=\left\{T_{1}^{I},T_{2}^{I},T_{3}^{I}\right\}\\ T^{W}=\left\{T_{1}^{W},T_{2}^{W},T_{3}^{W},T_{4}^{W}\right\} \end{array}\right.$$
(6)$$\left\{\begin{array}{c} M^{I}=\left\{M_{1}^{I},M_{2}^{I},M_{3}^{I}\right\}\\ M^{W}=\left\{M_{1}^{W},M_{2}^{W},M_{3}^{W},M_{4}^{W}\right\} \end{array}\right.$$

Then, the state transition function of the task can be expressed as follows:(7)$$\left\{\begin{array}{c} \sigma \left(S_{n}^{I},T_{n+1}^{I}\right)=\left(M_{n+1}^{I},S_{n+1}^{I}\right),n\in 0,1,2.\\ \sigma \left(S_{n}^{W},T_{n+1}^{W}\right)=\left(M_{n+1}^{W},S_{n+1}^{W}\right),n\in 0,1,2,3. \end{array}\right.$$

In the cable installation task, the mode of cooperation between SRL and the operator is as follows. (i) The operator arranges the cables (*PS*), and after finishing the arrangement, the switching condition $T_{1}^{I}$ is triggered by a voice command. The SRL grabs the cables near the operator’s hand ($M_{1}^{I}$). (ii) Task state enters *CS*. The SRL triggers $T_{2}^{I}$ through the state of the end jaws, tensioning the cable ($M_{2}^{I}$). (iii) The operator completes the cable installation with the assistance of the SRL grasping the cable (*IS*). The operator triggers $T_{3}^{I}$ through a voice command, and the SRL moves back to the standby position ($M_{3}^{I}$). (iv) The cable task ends. The operator and the SRL return to the initial state (*ES*), and the next cycle operation can be performed.

The cooperation mode of the electrical connector soldering task is as follows. (i) The operator prepares the soldering tools (*PS*) and triggers the switching condition $T_{1}^{W}$ through a voice command. The SRL grabs the electrical connector and enters a mode where the end can be dragged ($M_{1}^{W}$). (ii) The operator drags the end of the SRL near the soldering position (*DS*) and triggers $T_{2}^{W}$ by voice command. The end of the SRL automatically adjusts the appropriate soldering angle according to the posture of the soldering iron ($M_{2}^{W}$). (iii) The operator triggers $T_{3}^{W}$ by gazing after feeling that the posture is appropriate (*CS*). The SRL maintains the spatial position and pose of the tip ($M_{3}^{W}$). (iv) The operator operates with both hands to complete the soldering task (*SS*). The operator triggers the switching condition $T_{4}^{W}$ through a voice command, and the SRL returns to the initial position ($M_{4}^{W}$). (v) The soldering task is completed, and the operator and SRL can enter the next cycle operation (*ES*).

The task model based on *FSM* can realize the switching of the task state, which ensures the cooperation process between the operator and SRL. Cable installation tasks and electrical connector soldering tasks are both ambidextrous tasks. Therefore, proposing a task model for the operator and SRL cooperation is significant. Both task models are adaptable, which can be adapted to other cable installation tasks and soldering tasks through fine-tuning parameters.

#### 3.4.2. Switching Condition Determination and Task Parameter Estimation

To realize the task operation, after the task model based on *FSM* is constructed, it is necessary to determine the state switching conditions and related task parameters. In the cable installation task, the SRL’s task is to replace the task in the hands of the workers, clamping and stabilizing the cables. Therefore, our interaction scheme mainly invokes the hand joint and speech information. In Section 3.2, we analyzed the multi-source information of the human that can be obtained by the HoloLens2 device, including the speech input module and the hand joint recognition module. For the speech input module, we directly set the keywords (“Clamp” and “Come back”) required for the task. The state switching condition $T_{2}^{I}$ is determined by the SRL gripper’s opening and closing degree $E_{d}$. If $E_{d}$ is greater than the specified $E_{d0}$, it is considered that the clamping is successful, and $T_{2}^{I}$ is triggered.
(8)$$\left\{\begin{array}{c} \left\{T_{1}^{I},T_{3}^{I}\right\}=Keywords^{I}=\left\{“Clamp”,“Come\;back”\right\}\\ T_{2}^{I}=E_{d0} \end{array}\right.$$

We believe that the position information of the grasped object can be roughly estimated by the part of the hand between the thumb and the index finger when held by the human hand, so the position information of the four joints of the part of the hand between the thumb and the index finger ($P_{1}$, $P_{2}$, $P_{3}$, $P_{4}$) is extracted, as shown in Figure 8. Then, the grasping center position ($P_{0}$) can be expressed as the average value of the four joint positions under the action of the correlation coefficient.
(9)$$P_{0}=\left(P_{1}+P_{2}+P_{3}+P_{4}\right)/4$$

Therefore, the task state switching and SRL actions can be realized through the operator’s speech and hand joint information. The specific algorithm is shown in Algorithm 1.
**Algorithm 1***FSM* Task Model for Cable Installation Taskinitial valuesS←PS,S←NaN,M←NaN    WhileTruedo    Listen for speech keywords    $\;\mathrm{if}\;S=PS\;\mathrm{and}\;T=NaN\;\mathrm{and}\;{\mathrm{Keywords}}^{I}=“Clamp”\;\mathrm{then}$    $\;T\leftarrow T_{1}^{I}$    $\;\mathrm{Get}\;\mathrm{clamping}\;\mathrm{position}\;P_{0}\leftarrow \left(P_{1}+P_{2}+P_{3}+P_{4}\right)/4$    $\;\mathrm{SRL}\;\mathrm{Execution}\;\mathrm{Action}\;M\leftarrow M_{1}^{I}$    S←CS    $\;\mathrm{if}\;S=CS\;\mathrm{and}\;T=T_{1}^{I}\;\mathrm{and}\;E_{d}>E_{d0}\;\mathrm{then}$    $\;T\leftarrow T_{2}^{I}$    $\;\mathrm{SRL}\;\mathrm{switches}\;\mathrm{motion}\;\mathrm{mode}\;M\leftarrow M_{2}^{I}$    S←IS    $\;\mathrm{if}\;S=IS\;\mathrm{and}\;T=T_{2}^{I}\;\mathrm{and}\;{\mathrm{Keywords}}^{I}=“Comeback”\;\mathrm{then}$    $\;T\leftarrow T_{3}^{I}$    $\;\mathrm{SRL}\;\mathrm{returns}\;\mathrm{to}\;\mathrm{initial}\;\mathrm{position}\;M\leftarrow M_{3}^{I}$    S←ES    Break

The electrical connector soldering task mainly calls the operator’s speech, eye gaze information, and tracking method based on Vuforia. The keywords of state switching conditions $T_{1}^{W}$, $T_{2}^{W}$, $T_{4}^{W}$ are “Clamp”, “Adjust” and “Come back”. $T_{3}^{w}$ is triggered by the gaze target $Eye_{t}$. If the gaze point is on a preset virtual object $Eye_{obj}$, $T_{3}^{W}$ triggers.
(10)$$\left\{\begin{array}{c} \left\{T_{1}^{W},T_{2}^{W},T_{4}^{W}\right\}=Keywords^{W}=\left\{“Clamp”,“Adjust”,“Come\;back”\right\}\\ T_{3}^{W}=Eye_{t} \end{array}\right.$$

Vuforia is an augmented reality software development kit for mobile devices that can create augmented reality applications [31,32]. It uses computer vision technology to identify and track flat images and 3D objects in real time. This image registration capability enables developers to position and orient virtual objects, such as 3D models and other media, relative to real-world objects as they are viewed through the mobile device’s camera. The virtual object then tracks the position and orientation of the image in real time so that the observer’s viewpoint of view on the object corresponds to that of the target. Therefore, virtual objects appear to be part of the real-world scene.

To simplify the pose recognition of the soldering iron, Vuforia is used to identify the sign picture of its end, and the change in its position $P_{w}$ and pose $Q_{w}$ is tracked in real time, as shown in Figure 9. The end attitude of the SRL establishes a particular angular relationship with $Q_{w}$, making the soldering task more convenient. The specific state-switching method of the electrical connector soldering task is shown in Algorithm 2.
**Algorithm 2***FSM* Model of Electrical Connector Soldering Taskinitial valuesS←PS,S←NaN,M←NaN    WhileTruedo    Listen for speech keywords    $\;\mathrm{if}\;S=PS\;\mathrm{and}\;T=NaN\;\mathrm{and}\;{\mathrm{Keywords}}^{W}=“Clamp”\;\mathrm{then}$    $\;T\leftarrow T_{1}^{W}$    $\;\mathrm{SRL}\;\mathrm{enters}\;\mathrm{the}\;\mathrm{motion}\;\mathrm{mode}\;M\leftarrow M_{1}^{W}$    S←DS    $\;\mathrm{if}\;S=DS\;\mathrm{and}\;T=T_{1}^{W}\;\mathrm{and}\;{\mathrm{Keywords}}^{W}=“Adjust”\;\mathrm{then}$    $\;T\leftarrow T_{2}^{W}$    $\;\mathrm{Get}\;\mathrm{the}\;\mathrm{pose}\;\mathrm{of}\;\mathrm{the}\;\mathrm{soldering}\;\mathrm{iron}\;P_{w},Q_{w}$    $\;\mathrm{SRL}\;\mathrm{switches}\;\mathrm{motion}\;\mathrm{mode}\;M\leftarrow M_{2}^{W}$    S←CS    $\;\mathrm{if}\;S=CS\;\mathrm{and}\;T=T_{2}^{W}\;\mathrm{and}\;Eye_{t}=Eye_{obj}\;\mathrm{then}$    $\;T\leftarrow T_{3}^{W}$    $\;\mathrm{SRL}\;\mathrm{holds}\;\mathrm{end}\;\mathrm{position}\;\mathrm{and}\;\mathrm{attitude}\;M\leftarrow M_{3}^{W}$    S←SS    $\;\mathrm{if}\;S=SS\;\mathrm{and}\;T=T_{3}^{W}\;\mathrm{and}\;Keywords^{W}=“Come\;back”\;\mathrm{then}$    $\;T\leftarrow T_{4}^{W}$    $\;\mathrm{SRL}\;\mathrm{returns}\;\mathrm{to}\;\mathrm{initial}\;\mathrm{position}\;M\leftarrow M_{4}^{W}$    S←ES    Break

## 4. Experiments and Results

In this section, two typical multi-hand manipulation tasks are taken as examples to construct implementations and evaluate the augmented performance of humans.

### 4.1. Experimental Protocol

Before commencing experiments, it is essential to execute the equipment calibration procedure. Once the operator have donned and activated the SRL and HoloLens, their respective software applications seamlessly establish a wireless connection. Initially, the T265 camera acquires its initial coordinate system position, continuously updating it in real time to align with the movements of the SRL base. Upon the operator’s speech command of “calibrate”, the HoloLens program transitions into calibration mode. The operator gazes at the QR code affixed to the SRL until they receive the visual alignment display and the auditory cue, “Calibration completed”. Utilizing the methodology outlined in Section 3.3, the positional relationships among the coordinate systems of each system component are dynamically refreshed. The requisite system calibration preceding the experiment is successfully accomplished at this juncture.

A comparative experiment was carried out to analyze the augmentation effect of the SRL integrated with HoloLens on the wearer, as shown in Figure 10. The participants of these experiments were ten healthy men who had mastered the skills of cable installation and soldering. Two sets of comparative experiments were designed, including ten trials of wearing the equipment and ten trials of completing the task alone. The specific operation content is five times each of the cable installation task and the soldering task. A break of at least 30 min was inserted between each experiment. Data collection consisted of collecting recorded performance times. No personal data were collected during the experiment.

### 4.2. Results and Performance

The cable installation task focuses more on the cooperation of SRL and human hands to assist workers in clamping cables. The workers only need to complete the corresponding bundling tasks. SRLs can substantially reduce task difficulty and improve work efficiency. The electrical connector soldering task focuses on coordinating SRL with the tools. In this way, the operator can concentrate more on the soldering task at hand and reduce the difficulty of operation. Figure 11 and Figure 12 show the process of the participants wearing the HoloLens integrated SRL system to complete the cable installation task and electrical connector soldering task.

The average time for the ten participants to complete the task is shown in Figure 13. We separately performed statistical analyses on the data from the cable installation and electrical connector soldering tasks. We conducted a correlation analysis on the data about cable installation tasks, demonstrating the statistical significance of the data correlation (r = 0.284, sig = 0.045). A paired-sample t-test was also performed, revealing a statistically significant difference (t(49) = 23.751, *p* < 0.001) in the task completion time between the two conditions. The task completion time under the condition of wearing the device (M = 35.96, SD = 3.43) was significantly shorter than that under the condition of not wearing the device (M = 55.30, SD = 5.70). Similarly, we conducted a correlation analysis and a paired-sample t-test on the electrical connector welding task experiment data. The results demonstrated a significant correlation in the data (r = 0.396, sig = 0.004) and a statistically significant difference in task completion time between the two conditions (t(49) = 11.114; *p* < 0.001). The task completion time was significantly shorter when wearing the equipment (M = 39.92, SD = 8.36) compared to when not wearing the equipment (M = 71.12, SD = 21.62).

The results show that in two typical industrial tasks, the proposed system reduces the task completion time and improves the operator’s efficiency. The reduction in task time was more pronounced in soldering tasks. It may have something to do with how necessary the task requires a third arm. But its completion time is more unstable, and there is a phenomenon in which the task completion time after wearing the system exceeds that of completing the task alone. This may be related to different subjects’ soldering skills and equipment adaptability. The mean completion time of tasks shows that the system can improve the efficiency of human operations. It also demonstrates the usability of the equipment.

After completing the testing phase, we gathered qualitative feedback from participants regarding their subjective experiences. Participants reported experiencing heightened physical strain and indicated potential limitations in their capacity to utilize the system over prolonged durations. Some participants conveyed feelings of nervousness and apprehension stemming from their initial interactions and concerns regarding the safety implications associated with close proximity to the robotic arm. Furthermore, participants noted that the AR display obstructed their vision field, leading to operational task disruptions. The field of view provided by the HoloLens was found to be somewhat restricted, necessitating substantial head movements to adjust the viewing angle. Moreover, the calibration procedures carried out before the experiment were observed to be intricate and demanding in terms of effort.

These negative findings underscore the system’s limitations, encompassing physical and cognitive burdens, safety concerns, issues with obstructed line of sight, and operational complexities. We posit that alleviating psychological stress and enhancing safety could be achieved by incorporating additional holographic displays to provide prompt guidance for operations. Addressing the line-of-sight obstruction challenge entails combining gaze and spatial perception data to jointly optimize the positioning of holographic images, thus minimizing occlusions. Furthermore, simplification of the system calibration procedures is warranted.

## 5. Discussion

(1)This paper proposes an SRL interaction method based on Mixed Reality, which has the characteristics of intuitive display and easy development. This will provide new research ideas for the interaction of wearable robotic limbs. This paper also proposes an AR-based coordinate-tracking method and *FSM*-based task models. The task model is adaptable and can be used for similar tasks. Although the method based on Mixed Reality has an intuitive feedback function, its visual occlusion is also a problem that needs to be considered in future work.(2)The average task time under different experimental conditions was tested. The results indicate the possibility of system availability and enhanced human manipulation capabilities. But there is also some increased body load. This factor needs to be considered in designing both the structure of wearable robotic arms and interaction methods. The differences in the three-hand tasks will also affect the system’s ability to enhance the human body and the load index.(3)In the experiments of the cable installation task, the operation mode of replacing the human hand to hold the cable with the SRL and installing the cable is selected. Of course, it is also feasible to install cables on wearable robotic limbs and keep the cables by hand, but the task of cable installation has more subjective factors, and the operation is more complicated. Thus, this paper adopts a more straightforward collaboration strategy. We found that the ability of the wearable robotic limb to maintain the force and position of the end is essential, and it is used to ensure the tension and neatness of the cable, which can be used as a follow-up research direction.(4)In the experiments of the soldering task, a suitable angle was adopted to constrain the soldering angle between the electrical connector on the SRL and the end of the soldering iron to improve soldering comfort. The interaction research of wearable robotic limbs can pay more attention to the information of the human hand and the tools operated by the human hand. The wearable robotic limbs combine this information to construct strategies for interacting and collaborating with the wearer.

## 6. Conclusions

This paper proposes a novel human–robot interaction approach for wearable robotic limbs based on Mixed Reality and multimodal human information. We have constructed an interactive framework and a real-time tracking method for the system comprised of HoloLens and SRL. Task models based on *FSM* and interaction methods have been designed for two aircraft manufacturing tasks: cable installation and an electrical connector welding task. The results from the experiment involving ten participants suggest that the system enhances the operators’ task completion efficiency. However, some participants also reported experiencing physical and psychological burdens, obstructed vision, and operational complexity. This shows that the system has certain limitations. These negative aspects merit consideration in future work. The approach presented in this paper attempts to apply mixed reality to wearable robotic limbs, demonstrating initial potential for improving wearer efficiency. 

## Figures and Tables

**Figure 1 biomimetics-08-00479-f001:**
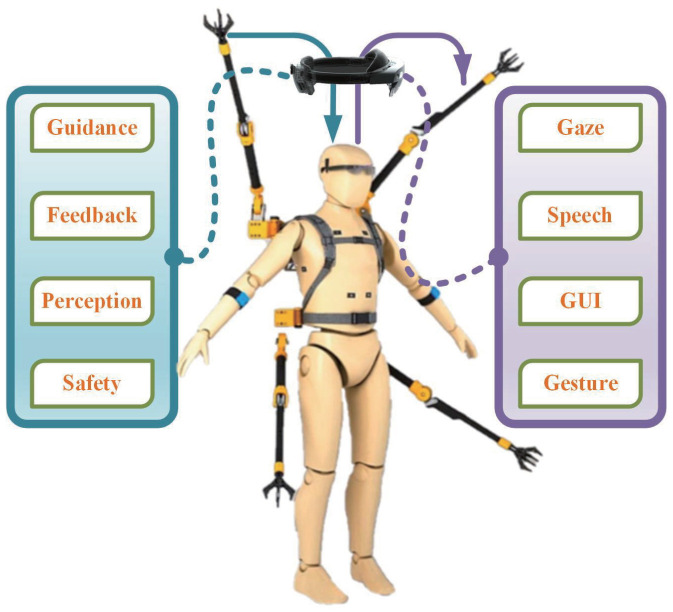
SRL system using HoloLens2 as the communication medium. The purple line represents the information flow from the human to SRLs, and the blue line represents the information flow from SRLs to the human.

**Figure 2 biomimetics-08-00479-f002:**
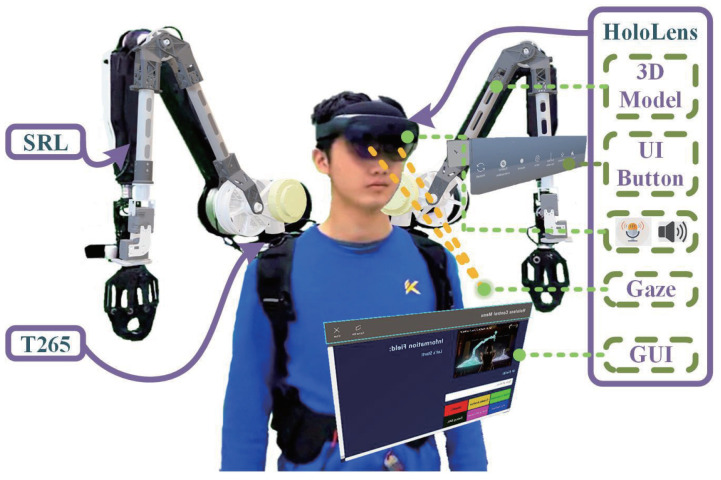
A diagram of the SRL system integrated with HoloLens. It also shows the application effect of some functions in HoloLens.

**Figure 3 biomimetics-08-00479-f003:**
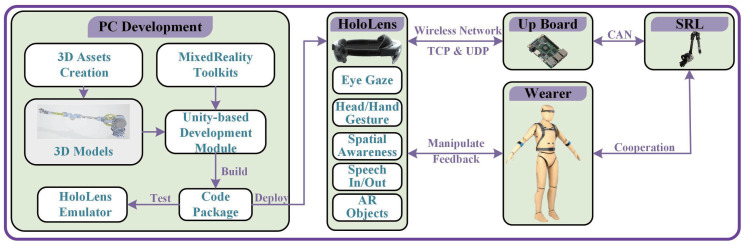
HoloLens2 program construction and integration.

**Figure 4 biomimetics-08-00479-f004:**
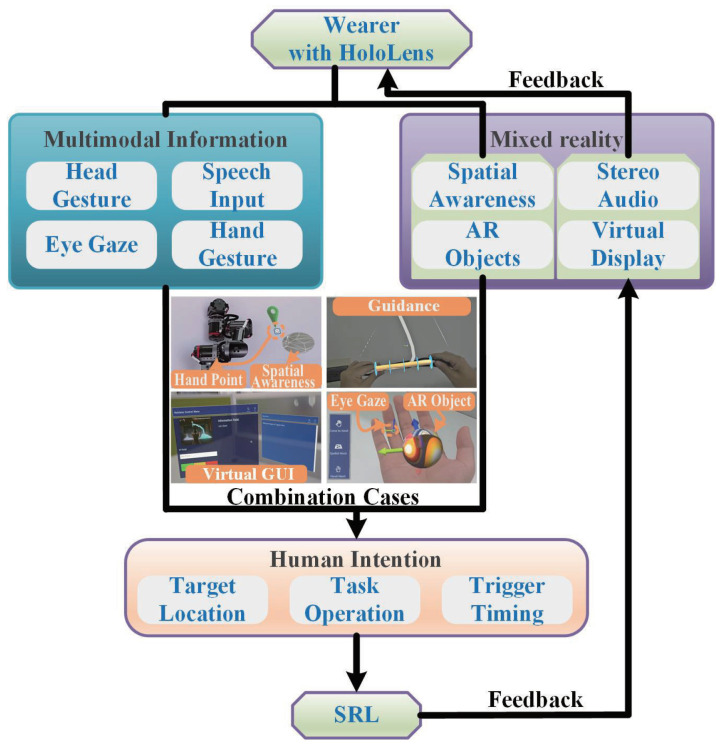
SRL interaction framework diagram based on mixed reality and human multimodal information. The blue box contains multimodal information about the human, and the purple box contains mixed reality content. The two are combined to obtain the three kinds of intent information in the orange box. The figure also shows several cases combining human body multimodal information and Mixed Reality.

**Figure 5 biomimetics-08-00479-f005:**
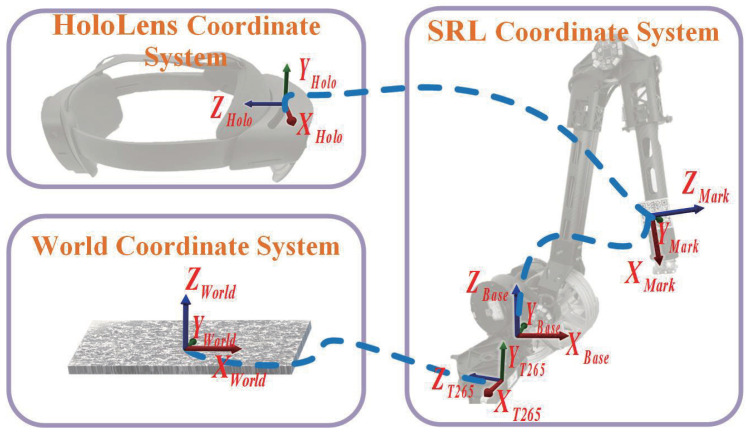
The coordinate system of HoloLens virtual space, world and the SRL real space.

**Figure 6 biomimetics-08-00479-f006:**
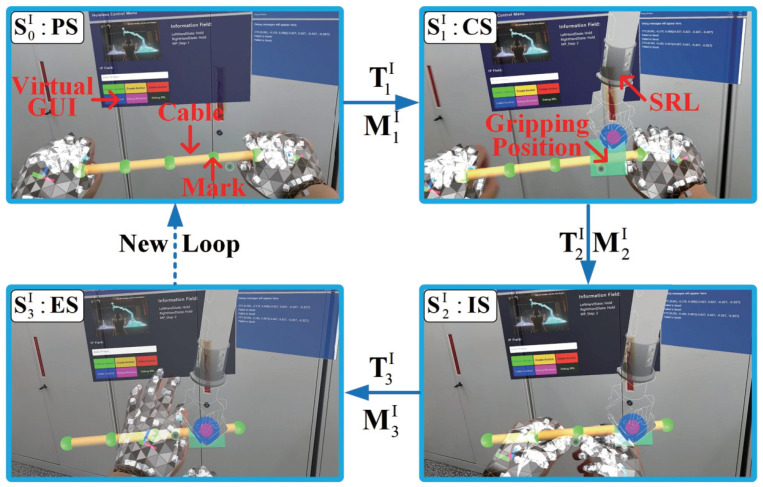
Description of the state transitions for the cable installation task. The view is the first-person perspective in HoloLens. The whole process is completed in the virtual environment, and the yellow cylinder represents the virtual cable with green installation marking points on it.

**Figure 7 biomimetics-08-00479-f007:**
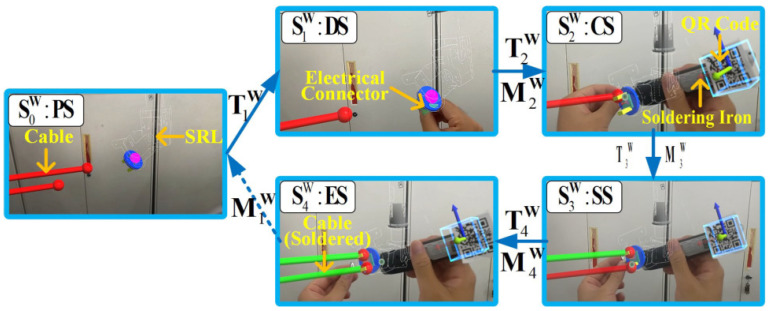
Description of state transitions for electrical connector soldering tasks. Red cylinders represent virtual cables. Electrical connectors are fixed at the end of the SRL model. The figure simulates the process of SRL assisting the operator to complete the welding task. When the electrical connector is successfully welded to the cable, the cable turns green.

**Figure 8 biomimetics-08-00479-f008:**
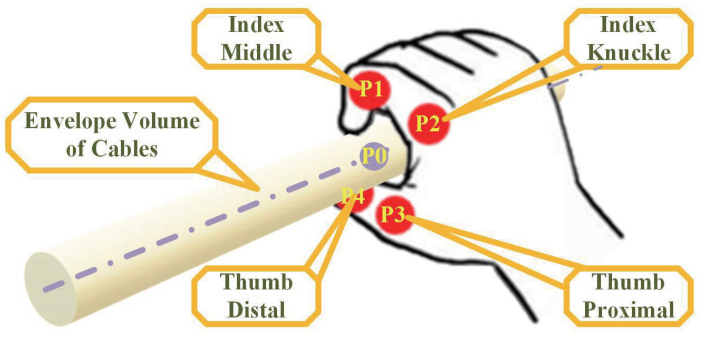
Schematic diagram of human hand joint information.

**Figure 9 biomimetics-08-00479-f009:**
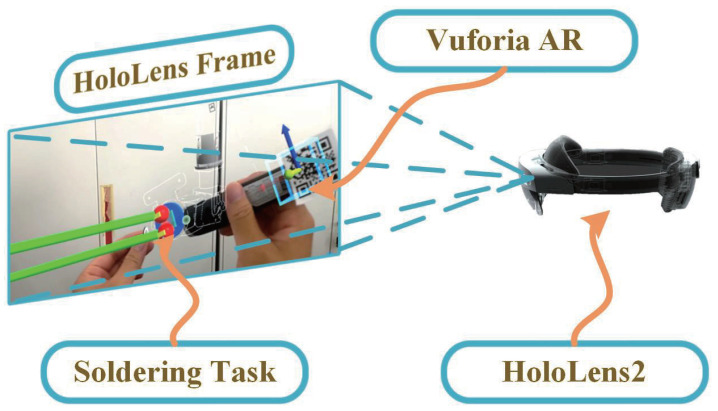
Application of Vuforia AR to realize position and posture tracking of the soldering iron.

**Figure 10 biomimetics-08-00479-f010:**
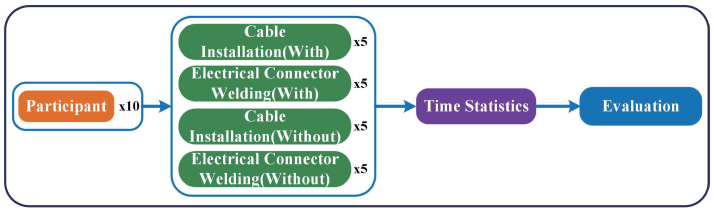
Organization Chart for Augmentation Effect Evaluation.

**Figure 11 biomimetics-08-00479-f011:**
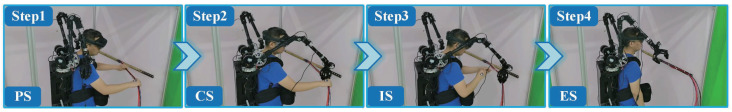
Procedure for cable installation tasks. Step 1: The operator organizes the cables. Step 2: The SRL grips and tensions the cable. Step 3: The cables are installed. Step 4: The mission ends, and the SRL returns to the preset position.

**Figure 12 biomimetics-08-00479-f012:**
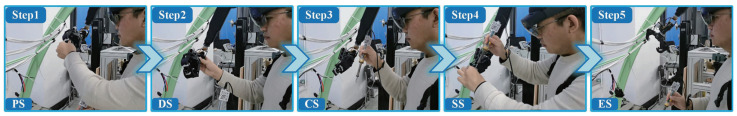
Experimental operating procedure for soldering tasks. Step 1: The SRL grabs the electrical connector and enters dragged motion mode. Step 2: The operator drags the end of the SRL to a suitable position. Step 3: The SRL end follows the soldering iron to adjust the angle. Step 4: The operator completes the soldering task. Step 5: The soldering task is completed, and the SRL returns to the preset position.

**Figure 13 biomimetics-08-00479-f013:**
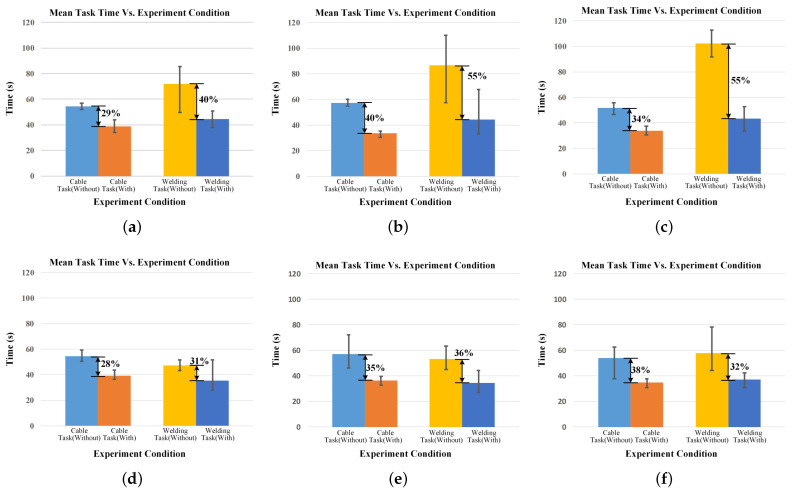
Mean task time of subjects under different experiment conditions. The figure also shows the percentage reduction in task time. (**a**) Mean task time for participant a. (**b**) Mean task time for participant b. (**c**) Mean task time for participant c. (**d**) Mean task time for participant d. (**e**) Mean task time for participant e. (**f**) Mean task time for participant f. (**g**) Mean task time for participant g. (**h**) Mean task time for participant h. (**i**) Mean task time for participant i. (**j**) Mean task time for participant j.

**Table 1 biomimetics-08-00479-t001:** Summary and classification table of related work.

Related Work	Classification
Interaction Methods of SRLs	Body Interface:
	(1) Interaction methods based on body motion mapping
	• Finger [10]
	• Feet [11]
	• Hand and finger force [12]
	(2) Interaction methods based on eye gaze [20,21,22]
	(3) Collaboration according to the operator’s actions [23]
	Muscle Interface—EMG:
	• Chest and abdominal EMG [13]
	• Forehead EMG [14]
	Neural Interface—EEG/MEG [15,16]
AR Visualization Methods	(1) Head-Mounted Displays (HMDs) [24,25,27,28]
	(2) Spatial Augmented Reality Projectors [25]
	(3) Hand-Held Displays (HHDs) [26]

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
