# Peer review of "Human Operation Augmentation through Wearable Robotic Limb Integrated with Mixed Reality Device"

_biomimetics, 2023, doi:10.3390/biomimetics8060479_

Round 1
Reviewer 1 Report
Hololens is ideal as an AytR tool, and a system that integrates it with robotics and utilizes it in a real place is considered to be effective. The process of building such a system and conducting user verification is worthy of praise. There is no problem with the structure of the paper.
There is no explanation about the robot's hardware. I also feel that it should written about problems caused by the hardware and the safety and disadvantages of installing it. When verifying with a specific task, it is desirable to have a number of subjects equivalent to the evaluation, but in the case of this paper, the number of subjects is small. Therefore, it is insufficient to generalize the effectiveness of the system. If the authors want to focus on the effectiveness of the system, a sufficient number of subjects are necessary.
Author Response
Dear Reviewer 1,
We would like to express our sincere gratitude for your thoughtful and constructive feedback on our manuscript titled "Human Operation Augmentation through Wearable Robotic Limb Integrated with Mixed Reality Device." Your valuable insights and suggestions have greatly contributed to the improvement of our work. We have carefully considered each of your comments and addressed them. The details are in the pdf attachment.
Sincerely,
Dr. Hongwei Jing

Reviewer 2 Report
The fact that I have no demands for changes may give you some feedback, without the need to violate the confidentiality of my evaluation.
Author Response
Dear Reviewer 2,
We would like to express our sincere gratitude for your feedback on our manuscript titled "Human Operation Augmentation through Wearable Robotic Limb Integrated with Mixed Reality Device." We have carefully considered your comments and addressed them. The details are in the pdf attachment.
Sincerely,
Dr. Hongwei Jing

Reviewer 3 Report
The topic of the work, which includes the application of virtual reality technologies in a variety of industries, from manufacturing to health area, is one that is becoming more and more popular. In this instance, a robotic arm linked to the human body is used in conjunction with VR technology to perform a variety of tasks. For interactive communication between the wearer and the robotic arm, a Hololens2 visualization system is used.
I would like to offer my thoughts on the work, which are presented here.
State of the art
The number of works examined looks about right. Numerous research have been conducted recently in this area. Instead of using the descriptive presentation from chapters 1 and 2, I advise you to create structured, tabular presentations based on certain clearly defined criteria.
Equipment
Realizing this system is a really good idea. It is noteworthy that the communication time is incredibly brief—20 ms.
It would be interesting to finish with a few technical details on the hardware and configuration of the robot (structure, weight, fixation method, prehensor...). All of these details are needed to get a sense of how complicated the system is and how wearable it is, as the equipment from FIG. 2 does not appear to be very simple to wear.
The equipment calibration method may need to be briefly explained.
The assignment given in section 3.4 is edifying, outlining the steps that must be taken.
You mentioned two different sorts of tests that were conducted by six men as part of the experiments. No information is provided regarding an endurance test that, in terms of duration, would be comparable to an operator's workday on an assembly line. It would be interesting to see how wearing and using such technology for a prolonged period of time affected human participants.
The concluding chapter has to be organized, perhaps with bullets for each concept. Additionally, it is necessary to present the test's negative findings.
Details
Strong colors should be used for text and diagrams in all figures in the article. The colors you used (light blue, light green, light purple, light orange) are not very visible, especially on a black/white print.
Author Response
Dear Reviewer 3,
We would like to express our sincere gratitude for your thoughtful and constructive feedback on our manuscript titled "Human Operation Augmentation through Wearable Robotic Limb Integrated with Mixed Reality Device." Your valuable insights and suggestions have greatly contributed to the improvement of our work. We have carefully considered each of your comments and addressed them. The details are in the pdf attachment.
Sincerely,
Dr. Hongwei Jing

Round 2
Reviewer 1 Report
As for the results of the subject test, there is a diagram comparing changes in task execution time, but aren't the factors inferred from the subjects' comments more important? Your paper will be better if you consider what disadvantages it has and what improvements need to be made.
Author Response
Dear Reviewer 1,
Thank you for your additional suggestions for this manuscript. Your suggestions have greatly inspired us. We have incorporated additional descriptions of the participants' subjective evaluations and summarized the limitations of the system and aspects that may require improvement. The changes are highlighted in red in the revised manuscript for your review.
Sincerely,
Dr. Hongwei Jing